# Recent Advances on Affibody- and DARPin-Conjugated Nanomaterials in Cancer Therapy

**DOI:** 10.3390/ijms24108680

**Published:** 2023-05-12

**Authors:** Federica Gabriele, Marta Palerma, Rodolfo Ippoliti, Francesco Angelucci, Giuseppina Pitari, Matteo Ardini

**Affiliations:** Department of Life, Health and Environmental Sciences, University of L’Aquila, 67100 L’Aquila, Italy

**Keywords:** affibodies, DARPins, cancer, therapy, targeting, delivery, nanoparticles, liposomes, proteins, DNA

## Abstract

Affibodies and designed ankyrin repeat proteins (DARPins) are synthetic proteins originally derived from the *Staphylococcus aureus* virulence factor protein A and the human ankyrin repeat proteins, respectively. The use of these molecules in healthcare has been recently proposed as they are endowed with biochemical and biophysical features heavily demanded to target and fight diseases, as they have a strong binding affinity, solubility, small size, multiple functionalization sites, biocompatibility, and are easy to produce; furthermore, impressive chemical and thermal stability can be achieved. especially when using affibodies. In this sense, several examples reporting on affibodies and DARPins conjugated to nanomaterials have been published, demonstrating their suitability and feasibility in nanomedicine for cancer therapy. This minireview provides a survey of the most recent studies describing affibody- and DARPin-conjugated zero-dimensional nanomaterials, including inorganic, organic, and biological nanoparticles, nanorods, quantum dots, liposomes, and protein- and DNA-based assemblies for targeted cancer therapy in vitro and in vivo.

## 1. Introduction

Therapies based on antibodies (Abs) are undoubtedly pivotal in several fields, including cancer treatments. Abs, especially monoclonals, have entered the mainstream for their use in the targeted delivery of chemotherapeutic agents and to manipulate anticancer immune responses. It is not surprising, therefore, that the number of approved Abs-based therapies is growing worldwide, propelling their clinical relevance [1,2,3]. In parallel, advances in nanotechnology to produce several types of nanomaterials have revolutionized nanomedicine for their small size, customizable surfaces, solubility, and biocompatibility, which make them able to interact with biological surfaces. For this purpose, biological macromolecules, particularly Abs, have been used as ligands to create advanced, smart hybrid nanomaterials addressed to therapeutic approaches [4].

However, although Abs exhibit strong binding and high selectivity toward the target epitopes and endless engineering possibilities, they also come with undesired drawbacks for applied purposes. Namely, Abs are large, bivalent, and multidomain proteins showing intramolecular oxidized cysteines forming disulfide bonds and often a glycosylation pattern. These features lead to relatively poor thermal and chemical stability. In addition, Abs only use the small complementarity-determining regions (CDRs) to interact with the antigen, and in some cases, the high cost of manufacturing at a large scale has been identified due to the complexity in producing the full Ab. This problem and the potential difficulty in penetrating solid tissues for cancer therapy justified the need for engineered derivatives with reduced size and composition [5].

In this context, other biological molecules with affinity properties toward ligands have been identified as valid alternatives to Abs. Nonantibody-binding proteins with low molecular weight have been identified and proposed as valuable tools and are currently being designed with improved properties. These antibody-mimicking molecules are grouped into two categories according to the location of the amino acid residues that mediate the binding to the ligands: those where the binding occurs via exposed, unstructured loops, and those where the interactions involve secondary structures, usually α-helices [5]. Among all, the so-called “affibodies” and “designed ankyrin repeat proteins (DARPins)”, both belonging to the second category, are the most representative of therapeutic means [1,3,6,7]. These affinity proteins have become invaluable tools in the development of next-generation therapeutics in vitro and in vivo for their unique biophysical and biochemical properties (see next paragraph), and their suitability in several applications is now well-established. For instance, affibodies can be easily designed with combined protein engineering approaches resulting in small and robust protein scaffolds showing favorable folding and stability. Moreover, affibodies encompass only 13 amino acid positions that differ between binding members and therefore, much of the knowledge to manipulate and functionalize these proteins is known [8,9]. A few examples can be recalled that highlight the importance of affibodies and DARPins in several biomedical applications that include both therapeutics [10,11,12] and imaging agents for tumors [13,14].

Considering such high versatility and biological activity, it is not surprising that nanomaterials have also been functionalized with both affibodies and DARPins to create hybrid and advanced structures for targeted delivery in vitro and in vivo [15]. Note, in some cases, they gave even more efficient results than immunoglobulins (Igs)-based approaches [15]. This review aims at collecting the most recent studies published in the last two years describing affibody- and DARPin-conjugated nanomaterials with zero-dimensional (0D) architecture, i.e., all three dimensions within 1–100 nm in size, including inorganic and organic nanoparticles, nanorods, and quantum dots as well as biological moieties such as liposomes, protein- and DNA-based assemblies for targeted cancer therapy in vitro and in vivo. The main features of affibodies and DARPins will be presented, as well as the strategies of synthesis and conjugation to nanomaterials and their efficiency on cells in vitro and tumors in vivo will be exposed.

## 2. Structural and Biochemical Features of Affibodies and DARPins

### 2.1. Affibodies

In 1984, the amino acid sequence of the virulence factor from *Staphylococcus aureus* called protein A (SpA) was published, unveiling five highly homologous domains A−E that encompassed 58 amino acid residues each [16]. These domains lacked cysteines and have been found to bind to Igs with high affinity [17,18,19]. The structural characterization revealed a simple bundle of three α-helices provided by nuclear magnetic resonance [20,21].

The SpA protein represents the precursor of affibodies, a new class of small, high-affinity Igs-binding proteins. The first affibody, named Z-domain, has been realized by mutating key amino acids of the SpA B-domain resulting in enhanced chemical stability and preserving the binding affinity. Furthermore, it showed enhanced resistance against low pH [22] and the typical native three-helix bundle [23,24] (Figure 1a). The Z-domain is a 58 amino acid molecule with an approximately 6.5 kDa molecular weight. It has been used to generate all known affibody libraries by combined mutations able to interact with various molecular targets. Examples of targets are Taq DNA polymerase, human insulin, and human apolipoprotein, showing K_D_ affinities in the µM range [25,26]. Moreover, affibodies targeting the epidermal growth factor receptor 2 (HER2), tumor necrosis factor α (TNFα), insulin and the platelet-derived growth factor receptor (PDGFR), and showing very high melting temperatures and K_D_ down to pM and fM were also realized [27,28]. Furthermore, other biochemical and biophysical aspects, such as the folding kinetics of the three-helix bundle, have been improved [29], thus contributing to enhancing their properties.

As a result, modern Z affibodies are small 58 amino acid polypeptides lacking cysteines and capable of rapid folding, which show high affinity for several molecular partners. Moreover, they can be easily engineered and expressed as soluble and proteolytically stable molecules in various host cells on their own or fused with other partners. These properties contributed to increasing the interest in affibodies for practical purposes, making them more appealing than Abs.

### 2.2. DARPins

Ankirin repeats (ARs) were discovered in the cell cycle regulators Swi6 from *Saccharomyces cerevisiae* and the cell division control protein Cdc10 and Notch from *Drosophila melanogaster* [32]. Since this discovery, ARs have been found in many eukaryotic proteins, becoming one of the most abundant repeat domains in the eukaryotic proteome alongside other repeats, i.e., leucine-rich repeats (LRR), armadillo repeats (ARM), and tetratricopeptide repeats (TPR). It is not surprising that more than 367,000 predicted AR domains have been found. Proteins with AR repeats show tightly packed tandem sequences of 4 to 6 repeats, which usually encompass 33 amino acids each. The repeats form a structural unit consisting of a β-turn followed by two antiparallel α-helices resulting in a typical helix−loop−helix−β-hairpin/loop structure. Short interdomain interactions stabilize a particular right-handed solenoid-like fold rather than a globular shape [33].

Similar to the original B-domain used to produce affibodies, the AR scaffold has been exploited to identify and randomize amino acids to manipulate the recognition properties, thus obtaining libraries of DARPins with an incredibly high yield of production (200 mg per liter of bacterial culture) and thermal stability [34]. Next-generation DARPins have been then produced by introducing a continuous convex paratope similar to the long CDR-H3 loop found in Igs without altering the biophysical properties of the original scaffold (Figure 1b). The resulting DARPins showed extended epitope-binding properties with affinity down to the pM range toward several targets, including human Igs, TNFα, the epidermal growth factor receptor (EGFR), and HER2 [35,36]. Further studies revealed that single point mutations strongly increased the thermal stability of these proteins up to melting temperatures of 90 °C [37].

Modern DARPins can recognize targets with specificities and affinities equal to or greater than Abs, disclosing a multitude of practical applications, including cancer therapies [38]. Furthermore, they can be produced with high yield through common bacterial expression systems, reaching high concentrations without aggregating, and show length-dependent stability against boiling and chemical denaturation.

## 3. Affibody- and DARPin-Conjugated Nanomaterials in Cancer Therapy

This paragraph collects recent studies reporting on affibody- or DARPin-conjugated 0D nanomaterials, that is, materials with all dimensions within the 1–100 nm range, for targeted therapy of cancer in vitro and in vivo. This survey shows that inorganic, organic, and biological nanostructures, e.g., metal and polymeric NPs, liposomes, and even whole viral capsids, can be successfully used, paving the way for multidisciplinary approaches and large-scale production. A brief overview describing the main conjugation strategies to obtain such hybrid bioconjugates will also be shown.

### 3.1. Inorganic Nanomaterials

Inorganic substances represent the evergreen materials to realize nanostructures with tailored size, shape, and features. As their use in cancer therapy is largely reported from applied to clinical studies [39,40], examples of conjugates containing affibodies or DARPins are increasing, thus demonstrating their usefulness as therapeutic and even imaging tools.

As a starting study, Shipunova et al. described the first photothermal therapy (PTT) taking advantage of metal silver NPs (AgNPs) conjugated to Z_HER2_, an affibody targeting HER2 that is overexpressed in cancer cells [41]. The authors have successfully synthesized plasmonic AgNPs with an approximately 35 nm diameter through green synthesis using aqueous extracts of *Lavandula angustifolia Mill* before decorating the metal surface by covalent crosslinking with several Z_HER2_ molecules. The hybrid Z_HER2_-AgNPs constructs have been tested in vitro on CHO and SKOV3-ip1 cells and in vivo in BALB/c Nu/Nu mice as thermosensitizers to induce local hyperthermia (Figure 2). Namely, the exposure of incident light at 465 nm was shown to induce localized surface plasmon resonance (LSPR) that locally raised the temperature to 42–47 °C leading to damage to cancer cells without affecting healthy cells. Another interesting aspect is that these constructs exhibit a wide range of temperatures upon excitation that strictly depends on their concentration and irradiation power, allowing the operator to vary the conditions for more effective PTT of cancers of different origins.

Though in a childhood stage, but like affibody-conjugated metal NPs as thermosensitizers, the PTT effect has been demonstrated elsewhere using DARPin-conjugated nanomaterials. In this regard, the example reported by Proshkina et al. is worth mentioning, for it reports DARPins covalently conjugated to gold nanorods (AuNRs) coated with bovine serum albumin (BSA) with length and width of approximately 50 and 8 nm, respectively. These hybrids have been synthesized by wet chemical reduction and used to treat HER2-overexpressing cancer cells [42]. Note, the researchers introduced the BSA coating to make the DARPin−AuNR constructs much more stable than the uncoated ones, strongly increasing their colloidal stability and biocompatibility in blood and the therapeutic effect, disclosing in vivo studies. The researchers demonstrated that the BSA-coated DARPins−AuNRs specifically accumulate in vitro in BT-474 and MDA-MB-231 cells and in vivo in BALB/c nude mice and that the irradiation of the tumor area with a laser at a 850 nm wavelength leads to about 70% inhibition of tumor growth in the treated mice.

Parallel studies have been carried out by Pourshohod et al. who demonstrated the versatility of AgNPs in a targeted therapeutic approach based on X-ray radiotherapy using Z_HER2−_AgNPs constructs [43]. In this study, AgNPs have been obtained by wet chemical reduction with a mean diameter of 120 nm and covalently conjugated to Z_HER2._ Their efficacy has been tested in vitro on various kinds of HER2-positive malignancies such as SK-BR-3, HN-5, and SKOV3 cells. An X-ray voltage of 6 MV has been applied on cells to be adsorbed by the metal surface of the hybrid constructs, which in turn undergo the emission of electrically charged particles through the photoelectric effect. In this way, the metal is used as a potent radiosensitizer to increase the radiation received by tumor cells while being blind and innocuous to normal cells.

As attractive alternatives to AgNPs, metal gold NPs (AuNPs) are extensively studied as biocompatible nanomaterials though they still lack huge interest as nanocarriers for targeted therapeutics against cancer. An interesting contribution to such an application is described by Zhang et al. who created complex Z_HER2_-DNA-AuNPs hybrid constructs for simultaneous delivery of 5-fluorodeoxyuridine (FUdR) and doxorubicin (Dox) to treat HER2-overexpressing cells [44]. The study shows the synthesis of about 31−39 nm large AuNPs by wet chemical reduction followed by covalently linking to FUdR-containing DNA strands carrying Z_HER2_ affibody molecules at the 3′-end, which maintain the binding ability toward the surface of cancer cells. The resulting coated AuNPs have been loaded with Dox molecules by intercalation into the DNA duplex regions and tested in vitro on human breast cancer cell lines BT474 and MCF-7. The constructs are shown to enter the cancer cells selectively and trigger intracellular Dnase II-mediated degradation of the exogenous DNA with consequent release of both FudR and Dox, leading to cell death. Moreover, the coloaded AuNPs showed a higher synergistic killing activity than the simple mixture of Dox and FudR.

A special mention of metal-based nanostructures is due to the constructs proposed by Roy et al., who used the so-called lanthanide-based upconverting NPs (UCNPs) conjugated to a hybrid affibody-cytosine deaminase (CD) prodrug-converting enzyme (ACD) to treat cancer cells overexpressing EGFRs [45]. In this study, neodymium (Nd)-, ytterbium (Yb)-, and thulium (Tm)-doped UCNPs have been synthesized using a thermal decomposition method leading to about 18 nm large NPs able to absorb near-infrared (NIR) light at 808 nm and emit ∼365 nm ultraviolet (UV) light. The UCNPs can be covalently crosslinked to fusion constructs expressing the benzophenone-modified protein couple Z_EGFR_-CD and tested in vitro on EGFR-overexpressing Caco-2 human colorectal cancer cells. Namely, the hybrid constructs Z_EGFR_-CD-UCNPs are photolinked under NIR) light to the exposed EGFRs and convert the administrated prodrug 5-fluorocytosine (5-FC) into the toxic anticancer compound 5-fluorouracil (5-FU) that kills cells. This effect is maintained in vivo upon administration of the constructs and the prodrug to BALB/c athymic nude mice bearing Caco-2 tumors showing a significant survival increase. An important aspect claimed by the authors is that the hybrid Z_EGFR−_CD-UCNPs can remain viable after a single covalent photoconjugation in vivo, which in turn localizes the targeted prodrug conversion for multiple weeks in a systemic manner. Note that this approach can also be exploited to realize advanced nanotools carrying drug cargos and contrast agents, thus enabling dual, combined effects for therapy and imaging both in vitro and in vivo [46].

Besides metal substances, materials with crystal architecture represent alternatives to gain interesting properties, being useful as scaffolds for bioconjugation and cancer therapy. An example is reported by Al-Ani et al. whose study describes a simple method to create lead sulfide quantum dots (PbSQDs) decorated with Afb2C affibodies and Zn(II)-protoporphyrin IX (ZnPP) to achieve theragnostic effects on HER2-positive cancer cells [47]. In this study, very thin crystals of PbSQDs have been made by wet chemical synthesis under basic conditions showing a diameter of about 5 nm and photoluminescence emission in the NIR wavelength and covalently conjugated to Z_HER2_ molecules, which in turn served as scaffolds to load the therapeutic agent, i.e., ZnPP covalently. The resulting ZnPP-loaded Z_HER2−_PbSQDs have been tested in vitro on SKBR3 breast cancer cells showing anticancer activity exclusively on HER2-positive cells and triggering apoptotic cell death.

The influence of the protein corona that affects many of the nanostructures administrated through the blood has been investigated by Ma et al. This example described an original approach where bacterial magnetosomes (BMPs) decorated with humanized antibody trastuzumab (TZ) via affibody and glutaraldehyde (GA) have been used to treat cancer cells while escaping the negative influence of the corona formed by plasma proteins [48]. The authors used the magnetotactic bacterium *Magnetospirillum gryphiswaldense* as a natural source of membrane-enveloped 30 to 120 nm large magnetite (Fe_3_O_4_) and greigite (Fe_3_S_4_) crystals forming the BMPs. The magnetosomes have been crosslinked to a HER2-targeting affibody and coated by adsorption of TZ, which is well known as a potent anticancer drug. The resulting Z_HER2_−TZ−BMPs hybrid constructs have been treated by incubation in normal human plasma and cancer cells or Igs-supplemented plasma, resulting in a small retention of extra proteins compared to bare BMPs and tested in vitro on the HER2-overexpressing breast cancer cell line SK-BR-3 for their targeting and uptake inside cells. Though lacking cell viability data, this study provided an original solution to achieve oriented adsorption of humanized therapeutic Abs, i.e., TZ, taking advantage of affibodies while discouraging the formation of a host corona on the Ab surface and maintaining affibody-driven targeting (Figure 3).

### 3.2. Organic Nanomaterials

Another huge area of applied nanomaterials in cancer therapy belongs to polymers and their nanostructured derivatives. There is currently a considerable interest in organic nanostructures as they could be designed for clinical applications while showing better biocompatibility and lower toxicity than inorganic ones [49].

A new drug delivery concept is currently being explored based on a two-step strategy to outperform the traditional one-step delivery for HER2-overexpressing cells significantly. In this context, the example reported by Komedchikova et al. is worth mentioning as it exploits a pretargeting step that increases the affinity of a second NP-linked cytotoxic compound [50]. In this study, the authors developed a first targeting module consisting of an anti-HER2 polypeptide DARPin fused with the protein barstar, which is known to exhibit an extremely high affinity in binding to the ribonuclease barnase, i.e., the barnase−barstar interaction is one of the strongest discovered. As a second module, poly-(D,L-lactic-co-glycolic acid) NPs (PLGANPs) with about 218 nm diameter are chemically synthesized by double emulsion “water−oil−water” method, loaded with the diagnostic Nile Blue dye and the therapeutic drug Dox before covalently linking to the barnase enzyme (Figure 4). The system has been tested in vitro on human breast adenocarcinoma cells SK-BR-3 and Chinese hamster ovary cells CHO. The assays have been carried out by sequential administration of the DARPin−barstar fusion protein followed by the therapeutic hybrid complex PLGANPs−Dox−barnase, revealing that this strategy decreases 100 times the effective amount of the therapeutic nanocarrier to kill cells compared to the single-step treatment. The authors highlighted the importance of using the barnase−barstar system for its unique stability under severe conditions in terms of low pH, high temperatures, and the presence of chaotropic agents and for its absence in mammals, making this complex very stable within the bloodstream without any interaction with the endogenous host components. Moreover, this example highlights the use of a first nontoxic module, i.e., DARPin−barstar, and subsequent targeting with a second complementary toxic module, which is the PLGANPs−Dox−barnase construct, as a solution for decreasing doses for administration and lowering systemic toxicity.

A bioconjugated construct with a therapeutic effect is shown by Shipunova et al. who developed PLGANPs loaded with a fluorescent photosensitive xanthene dye, i.e., Rose Bengal, and linked to the Z_HER2_ affibody for targeting and killing HER2-overexpressing cells [51]. In this study, PLGA has been polymerized by a “water-in-oil-in-water” emulsion method in the presence of dye and coated with chitosan oligosaccharide to form biocompatible and biodegradable PLGANPs with an average diameter of 120 nm. The resulting NPs have been then covalently conjugated to Z_HER2,_ resulting in a hybrid construct loaded with Rose Bengal, the latter exploited as both a fluorescent dye and toxic reagent capable of producing reactive oxygen species (ROS) upon irradiation. The researchers showed that the chemical conjugation of PLGA with an anti-HER2 affibody resulted in the selective binding to CHO and SK-BR-3 cancer cells in vitro. The irradiation with a green light at 540 nm caused cancer cell death. The authors also claimed that the components of such a hybrid polymeric delivery system are already approved for clinical applications, thus suggesting a rapid translation into clinical practice for diagnostic and therapeutic applications.

Zhang et al. used a peptide-based polymer fused to the Z_HER2_ affibody and loaded with FUdR as a therapeutic agent to target and kill gastric cancer cells [52]. In this example, the authors produced by bacterial heterologous protein expression the Z_HER2_ fused to an amphipathic 30 amino acid peptide known as RALA that contains 7 arginines, facilitating condensation of nucleic acids, and 6 leucines that are known to help the peptide to traverse cell membranes. The fused construct can electrostatically bind to negatively charged oligomeric strands of FUdR and assemble into Z_HEr2_−FUdR−RALANPs with a mean diameter of 104.5 nm. The construct has been tested in vitro, showing excellent targeting and higher cytotoxicity in HER2-overexpressing gastric cancer N87 cells compared to the free FUdR. Moreover, the researchers revealed that the anticancer mechanism of the constructs enhances the expression and activity of the apoptosis-associated proteins Bcl-2/Bax-caspase 8,9-caspase 3 apoptotic pathway induced by FUdR.

Similar studies have taken advantage of self-assembling short polymers forming micellar drug carriers in an aqueous solution. This is the case of two parallel studies where the Z_HER2_-conjugated anticancer drug monomethyl-auristatin E (MMAE), able to form nanomicelles, has been used to treat ovarian cells, breast cells, and colorectal cancer cells. In one example, Xia et al. produced a cysteine-modified Z_HER2_ affibody covalently attached to MMAE via a cleavable linker containing a cathepsin B-responsive valine−citrulline dipeptide and a para-amino-benzyloxycarbonyl (PABC) spacer [53]. The authors used the hybrid, self-assembling 153 nm Z_HER2_-MMAE nanomicelles to treat in vitro HER2-overexpressing ovary and breast cancer cells SKOV-3, BT474, and MDA-MB-231 and in vivo tumor-bearing mice. The constructs revealed an increased circulation time in blood and enhanced tumor targeting capacity and drug accumulation with important results of tumor eradication in both small and large solid cell masses of about 100 and 500 mm^3^, respectively (Figure 5). The mechanism underlying the MMAE release has been shown to depend on the cleavage by the cathepsin B enzyme upon cellular internalization. A similar example exploiting the same nanomicellar construct reported by Yang et. al. showed an affibody moiety directed to the PDGFRβ, a major vascular target identified in several tumors [54]. The authors observed that the Z_PDGFRβ_−MMAE constructs formed 130 nm large nanomicelles, which can target and kill in vitro COLO 205, HCT-116, and LS174T colorectal cancer cells and in vivo tumor-bearing BALB/c nude mice and SD rats with a tumor inhibition rate reaching over 99%.

Another strategy paving the way for modern chemotherapy is based on dual targeting that makes possible improvements in both efficiency and specificity while minimizing side effects. Also known as “image-guided targeted delivery”, such a strategy is considered a promising approach to reach combined cancer theranostics, especially against aggressive tumors. A good example of the dual targeting system has been reported by Shipunova et al. who developed a complex formulation including affibodies, DARPins, and PLGANPs loaded with Dox for synergistic immune/chemotherapy of HER2-overexpressing cancer cells [55]. In this study, 140 nm PLGANPs synthesized by the “water−oil−water” method, loaded with Nile Red fluorescent dye and Dox have been covalently linked to HER2-targeting Z_HER2_. Furthermore, an immunotoxin containing the HER2-targeting DAPRin fused to therapeutic *Pseudomonas exotoxin* A (LoPE) has been made by heterologous expression. The constructs Z_HER2_−Dox−PLGANPs and DARPin-LoPE have been used to target subdomains III and IV and the subdomain I of HER2, respectively, in vitro on human breast adenocarcinoma SK-BR-3, ductal carcinoma BT-474, lung carcinoma A549, and Chinese hamster ovary CHO cancer cells and in vivo in tumor-bearing BALB/c nude mice. The authors reached 100% cell death and a 1000-fold reduction of the effective concentration of the DARPin−LoPE immunotoxin to achieve the same therapeutic effect using the monotherapy. Furthermore, the strategy leads to the prevention of secondary tumor nodes. This example also recalls an important aspect concerning therapeutics against cancer: toxins, indeed, are currently being investigated as effective tools for their high potency and ease of production though clinical use is still missing [56].

### 3.3. Hybrid Nanomaterials

As the inorganic and organic materials are combined to create nanometric composites for practical applications, new chemical and/or physical features also arise to be exploited in nanomedicine. Though the use of these hybrid nanostructures is primarily known in the field of tissue regeneration, their applications in cancer theragnostics are rising quickly [57,58].

An interesting example is shown by Wang et al. reporting on multifunctional, cisplatin-loaded mesoporous NPs, including polydopamine (PDA) and MnO_2_ and linked to dimeric Z_HER2_ to treat HER2-positive ovarian tumors [59]. The PDANPs have been made by a nano emulsion assembly and loaded by electrostatic forces with the positively charged cisplatin molecules before coating with a thin MnO_2_ layer by in situ reduction and further coated with a PDA layer to increase the biocompatibility in cells. Dimeric Z_HER2_ molecules expressed in *Escherichia coli* have been linked covalently to the PDA layer to result in multifunctional and multicomponent Z_HER2_−cisplatin−PDA−MnO_2−_PDANPs with an average size of 163 nm. The authors tested these nanocomposites in vitro on HER2-positive ovarian cancer SKOV-3 and MCF-7 cells and in vivo in tumor-bearing BALB/C nude mice for their dual-mode therapeutic role (Figure 6). Indeed, the NPs provided a targeted cytotoxic effect due to the cisplatin cargo and resulted as radiosensitizers upon irradiation with X-ray causing the formation of ROS species upon decomposition of H_2_O_2_. It is important to recall that these nanocarriers are biodegradable under a tumor microenvironment while also providing a means to realize imaging for the tumor target cells, thus being very versatile affibody-driven nanotools for cancer theragnostics.

An alternative approach is shown by Novoselova et al. who demonstrated the therapeutic effect of HER2-targeting nanocomposites on cancer cells upon remote release of cytostatic Dox by low-intensity focused ultrasound (LIFU) [60]. In this study, Dox and magnetite NPs have been encapsulated by freezing-induced loading before coating with a polymeric shell bearing a couple of bilayers made by polyarginine and dextran sulfate resulting in 400 nm NPs. The carboxyl groups of the dextran component have been used to covalently immobilize HER2-targeting DARPin molecules, and the resulting hybrids tested both in vitro on ovarian adenocarcinoma HER2-overexpressing SKOV-3 cells and in vivo on tumor-bearing BALB/c nude mice. The researchers demonstrated that these hybrid Dox-loaded magnetic NPs efficiently targeted HER2-overexpressing cells and that the application of controlled LIFU led to remote destruction of the carriers. Furthermore, they exhibited controlled release of the therapeutic Dox, killing the cancer cells while not affecting healthy cells. Interestingly, the data provided hints highlighting “stealth behavior” nanocomposites to bind to mitochondria and escape internalization inside late endosomes or lysosomes. These results open the possibility of using these nanocomposites to overcome drug resistance and interact with mitochondria, which are among the most important subcellular targets for delivering therapeutic drugs.

### 3.4. Biological Nanomaterials

Biological supramolecular assemblies are naturally occurring nanoscopic objects with desired properties in nanotechnology, such as regular shape, binding specificity, and affinity, solvent-exposed surfaces for functionalization, and catalytic activity [61]. Accordingly, biomolecules have been widely exploited to create smart and/or functionalize nanostructures with multiple combinations, thus enabling tailored creation of hybrid bioconjugates at the nanoscale for different purposes [62,63,64,65].

In such a huge context, a flourishing area encompasses biological nanostructures, including DNA and protein NPs, liposomes, and even whole capsids. For example, Jia et al. demonstrated how liposomes loaded with Dox and coupled to Z_EGFR_ selectively targeted and produced high cytotoxicity on EGFR-overexpressing cells [66]. In this study, PEGylated liposomes with diameters of 110–137 nm have been obtained chemically by the ethanol injection method from hydrogenated soybean phosphatidylcholine, cholesterol, and methoxypolyethelene glycol distearyl-phosphatidyl-ethanolamine (mPEG 2000-DSPE). Dox molecules have been encapsulated through an active loading method using ammonium sulfate. The Dox-loaded liposomes have been linked covalently to Z_EGFR_ affibodies, and the resulting constructs tested in vitro on human epidermoid carcinoma A431 cells and mouse melanoma B16F10 cells and in vivo on tumor-bearing BALB/c (nu/nu) nude mice. The results showed that the Z_EGFR_ affibody linked to the Dox-loaded PEGylated liposomes is highly efficient compared to non-targeted liposomes, improving the therapeutic effect in mice inoculated with cell lines. Furthermore, the in vivo studies revealed long circulation time and efficient accumulation in tumors and, notably, reduced systemic toxicity such as body weight loss and organ injury. The same authors exploited this strategy to treat cancer cells by liposome carriers loaded with cisplatin [67]. In this case, 140 nm large, PEGylated liposomes linked to Z_EGFR_ have been used in vitro and in vivo with remarkable therapeutic results.

A special mention is due to toxins as therapeutic tools in cancer treatment. Biological toxic agents, such as protein domains or gene sequences, have been explored during the last decades to deliver agents into cancer cells. However, they still are subjected to two main strategies based on Ab-mediated toxin delivery or immunotoxins or the use of genes coding for toxic proteins or nonhuman enzymes, for instance, those from bacteria or plants [56,68]. Taking advantage of DARPin scaffolds, new strategies have been developed as reported, for instance, by Shipunova et al. who proposed a targeted delivery of the bacterial toxin ribonuclease barnase from *Bacillus amyloliquefaciens* into HER2-positive human cancer cells using liposomes [69]. Liposomes with a mean diameter of 117 nm have been assembled from L-α-phosphatidylcholine and phosphatidylethanolamine and loaded with barnase by electrostatic adsorption accounting for the protein-positive charge and the negative inner surface of the liposomal micelles. The outer surface of the barnase-loaded liposomes has been then functionalized with HER2-specific DARPins by covalently linking and tested in vitro on SK-BR-3 cells as adenocarcinoma cancer cell lines. This study demonstrated for the first time the possibility of targeting and killing HER2-specific cancer cells with a bacterial toxin taking advantage of a DARPin scaffold.

When speaking about biological nanostructures, proteins and their macromolecular complexes are at the forefront in nanotechnology-based therapies. Proteins naturally undergo self-assembly into regular, sharp structures with multiple chemical groups to functionalize while exhibiting regular shape and enzymatic activities [61]. In therapy, proteins are highly desired as they can be loaded with multiple moieties and produced with high yield, purity, and cost-effective methods. One interesting example comes from Jun et al. who used protein cage-like NPs of lumazine synthase isolated from *Aquifex aeolicus* (*Aa*LsNPs) to treat cancer cells overexpressing EGFR and loaded with the TNF-related apoptosis-inducing ligand (TRAIL), a promising anticancer agent, taking advantage of affibodies [70]. The authors produced an engineered form of *Aa*LsNPs carrying the 13-amino acids peptide sequence ST and an engineered form of the TRAIL and EGFR-targeting affibodies carrying the 13 kDa peptide SC, both peptides derived from the second immunoglobulin-like collagen adhesin domain of *Streptococcus pyogenes*. The engineered protein *Aa*Ls still exhibits the ability to self-assemble into the native hollow icosahedral capsid architecture with a mean diameter of 40 nm even after functionalization with the engineered TRAIL and affibodies. The functionalization occurs through the spontaneous formation of irreversible isopeptide covalent bond that forms between ST and SC. The Z_EGFR_–TRAIL–*Aa*LsNPs constructs have been assessed in vitro on EGFR-overexpressing A431 cancer cells, effectively disrupting the EGF-mediated signaling pathway as well as strongly activating both the extrinsic and intrinsic apoptotic pathways (Figure 7). The therapeutic effect has also been tested in vivo on tumor-bearing mice, which exhibited a noticeable suppression of tumor growth, thus confirming the synergistic antitumor efficacy of the hybrid construct. This example demonstrates how to apply an effective protein-based therapeutic approach and highlights the versatility of proteins to ease surface functionalization for carrying multiple protein-based ligands and modulators.

A new class of biological nanostructures is represented by tetrahedral DNA structures. Tetrahedral DNA nanostructures are three-dimensional pyramidal complexes of four complementary single strands, which have been proposed as drug carriers for their high stability, biocompatibility, suitability for different drugs, and cellular uptake rates [71]. An example is the study by Zhang et al. who used these nanostructures to treat HER2-positive breast cancer, taking advantage of HER2-directed affibodies [72]. Four DNA single strands have been generated by solid-state synthesis and functionalized with four polymeric FUdR strands. One strand has been further covalently linked to the Z_HER2_ affibody and left to self-assemble with the remaining three strands to form the final DNA tetrahedral nanostructures with pyramidal architecture and mean size of about 23 nm. The chimeric Z_HER2_–FUdR–DNA tetrahedrons have been assessed in vitro on breast cancer cells BT474 and in vivo on tumor-bearing mice revealing long blood circulation, specific accumulation in the tumor region, and relevant antitumor efficacy line (Figure 8). The same authors provided another example of a DNA nanostructure for cancer therapy where an affibody-modified G-quadruplex DNA micelle was coloaded with FUdR and curcumin (Cur) for the treatment of HER2-positive gastric carcinoma cells [73]. Cur molecules were trapped within the hydrophobic core of the Z_HER2_-FUdR-DNA micelle resulting in about 132 nm constructs, which have been tested as anticancer agents in vitro on HER2-positive N87 gastric cells revealing an anticancer mechanism based on FUdR-induced Bcl-2/Bax-caspase 8,9-caspase 3 apoptosis pathway.

A comprehensive overview of the aforementioned affibody- and DARPin-conjugated 0D nanomaterials, their main features, and applications are reported in Table 1.

### 3.5. Conjugation Strategies

As for many bioconjugated hybrid nanostructures, the constructs reported in this review are primarily obtained by two main strategies, namely (1) chemical covalent conjugation through crosslinkers and (2) bio-inspired covalent conjugation. A brief description is recalled herein, and a comprehensive overview is provided in Table 2, describing main properties in terms of constitutive matter, shape, and conjugation strategy.

First and foremost, covalent chemical crosslinking of affibodies and/or DARPins to nanomaterials represents the most common strategy to achieve unaggregated and monodisperse hybrids with chemical and thermal stability under several conditions, in vitro and in vivo, inside and outside cells. In addition, chemical crosslinking ensures that both the biological and nonbiological components are codelivered to the same target and achieve the desired therapeutic effect. Chemical covalent crosslinking between affibody/DARPins and nanomaterials is usually achieved by taking advantage of solvent-exposed chemical groups, e.g., carboxylate, ammine, and thiol, which are located on the surface of both the nanomaterial and the protein; these groups are ready-to-use anchoring sites to allow crosslinkers to bridge. Amongst the existing crosslinkers, common reagents to produce affibody/DARPin-conjugated nanostructures can be found as 1-ethyl-3-(3-dimethyl-aminopropyl) carbodiimide (EDC), N-hydroxysuccinimide ester (NHS), N-ε-malemidocaproyl-oxysuccinimide ester (EMCS), N-succinimidyl 3-(2-pyridyldithio)propionate) (SPDP), PEG-distearoylphosphatidylethanolamine (PEG-DSPE), and their derivatives. Note, these molecules show good solubility in aqueous solutions allowing the conjugation between the nanomaterial and the affibody/DARPin to occur efficiently without organic solvents and quickly [74].

Bio-inspired conjugation is also another strategy to obtain affibody- and DARPin-conjugated nanomaterials. In the context of this review, special mentions are due to the protease-cleavable valine-citrulline linker and the ST-SC spontaneous isopeptide. Valine-citrulline is commonly used to obtain many drug delivery systems, primarily antibody-drug conjugates (ADC), and is now under consideration to link affibodies and DARPins to nanomaterials too. This system, however, shows suboptimal stability in vivo, causing adverse effects and leading to clinical issues. For these reasons, derivatives such as glutamic acid-glycine-citrulline (EGCit) linkers have been produced to solve the problem [75]. The spontaneous bond that forms within the ST-SC isopeptide is another interesting example concerning the production of therapeutics. This isopeptide is a stabilizing posttranslational modification found in Gram-positive bacteria and can be reproduced under mild conditions. This is covalent, irreversible, and as stable as an amide bond under broad pH ranges and temperatures, several redox environments, and against nonionic detergents [76].

An interesting aspect concerning the conjugation correlates with the number of affibodies or DARPins per NP and its effect on the targeting efficiency. A study has been recently reported to compare affibodies, DARPins, and full-length Abs linked to 20 nm magnetic NPs and their targeting efficacy on HER2-overexpressing cells [15]. This study showed that the number of DARPins and affibodies bound to the surface of NPs is 10 and 40 times, respectively, higher than the number of Ab molecules, likely because of the significant difference in size. Furthermore, the higher loading amount is also shown to increase the delivery efficiency into the cells. The authors explained these results in terms of avidity. The HER2 receptor forms clusters which are associated with lipid rafts on the cell membrane; NPs exhibiting high ligand density, i.e., those carrying affibodies or DARPins, are much more prone to multipoint binding toward the receptors, thus leading to a higher avidity of such interactions. Conversely, the lower ligand density of the Abs-conjugated NPs might lead to binding with 1:1 stoichiometric ratio and therefore decreased targeting. Therefore, unless steric factors occur, the conjugation strategy to achieve a high number of biomolecules linked to the NPs should be favored when using affibodies or DARPins.

Note, it must be recalled that in some cases, an intermediate density of affibodies on NPs can provide significant improvements in cell binding in comparison with higher and lower ligand amounts. This effect has been observed across NPs with different hydrodynamic diameters and cells with low receptor densities [77].

**Table 2 ijms-24-08680-t002:** Main features of the affibody- and DARPin-conjugated nanomaterials cited in this review.

Inorganic Nanomaterials					
Material	Synthesis	Shape	Size ^1^	Bioconjugation Strategy	Reference
Ag	Biological synthesis	Particle	35 nm	Crosslinking with EDC/NHS	[41]
Au	Chemical synthesis	Rod	50 × 8 nm	Crosslinking with 2-iminothiolane hydrochloride and sulfo-EMCS	[42]
Ag	Chemical synthesis	Particle	120 nm	Crosslinking with sulfo-SMCC or EDAC/NHS	[43]
Au	Chemical synthesis	Particle	31–39 nm	Crosslinking with sulfo-EMCS	[44]
Nd, Yb and Tm	Chemical synthesis	Particle	18 nm	Crosslinking with NHS-PEG-azide	[45]
Pb, S	Chemical synthesis	Dot	5 nm	Crosslinking with EDC/NHS	[47]
Fe_3_O_4_, Fe_3_S_4_	Biological synthesis	Particle	30–120 nm	Crosslinking with SPDP	[48]
**Organic Nanomaterials**					
**Material**	**Synthesis**	**Shape**	**Size ^1^**	**Bioconjugation to Affibody/DARPin**	**Reference**
PLGA	Chemical synthesis	Particle	120 nm	Crosslinking with EDC/NHS	[51]
RALA	Biological synthesis	Particle	104.5 nm	Fusion synthesis	[52]
MMAE	Chemical synthesis	Micelle	153 nm	Crosslinking with valine-citrulline dipeptide and PABC spacer	[53]
MMAE	Chemical synthesis	Micelle	130 nm	Crosslinking with valine-citrulline dipeptide and PABC spacer	[54]
PLGA	Chemical synthesis	Particle	218 nm	Fusion synthesis; protein-protein high affinity interaction	[50]
PLGA	Chemical synthesis	Particle	140 nm	Fusion synthesis; crosslinking with EDC/NHS	[55]
**Hybrid Nanomaterials**					
**Material**	**Synthesis**	**Shape**	**Size ^1^**	**Bioconjugation to Affibody/DARPin**	**Reference**
PDA, MnO_2_	Chemical synthesis	Particle	163 nm	Fusion synthesis; crosslinking with Michael addition/Schiff base reaction	[59]
CaCO_3_, Fe_3_O_4_, polyarginine, dextran sulfate	Chemical synthesis	Particle	400 nm	Crosslinking with EDC/NHS	[60]
**Biological Nanomaterials**					
**Material**	**Synthesis**	**Shape**	**Size ^1^**	**Bioconjugation to Affibody/DARPin**	**Reference**
Hydrogenated soybean phosphatidylcholine, cholesterol and mPEG 2000-DSPE	Chemical synthesis	Micelle	110–137 nm	Crosslinking with maleimide-PEG DSPE	[66]
Hydrogenated soybean phosphatidylcholine, cholesterol, and mPEG 2000-DSPE	Chemical synthesis	Micelle	140 nm	Crosslinking with maleimide-PEG DSPE	[67]
L-α-phosphatidylcholine and phosphatidylethanolamine	Chemical synthesis	Micelle	117 nm	Crosslinking with 2-iminothiolane hydrochloride and sulfo-EMCS	[69]
*Aa*LS protein	Biological synthesis	Particle	40 nm	Crosslinking through spontaneous ST-SC isopeptide covalent bond	[70]
DNA	Chemical synthesis	Tetrahedron	23 nm	Crosslinking with EMCS	[72]
DNA	Chemical synthesis	Micelle	132 nm	Crosslinking with EMCS	[73]

^1^ Average size.

## 4. Conclusions

Affibody- and DARPin-based therapeutics are quickly becoming valuable tools in cancer treatments. It is known, indeed, that some compounds based on these proteins are already in clinical trials for both imaging and therapeutic purposes. Namely, DARPins are being used as therapeutics and affibodies mainly restricted to imaging of tumors [5]. However, the first affibody-based agent named Izokibep (ABY-035), made by the company Affibody Medical AB, recently entered phase II clinical trials as an interleukin-17A inhibitor against psoriatic arthritis. These advances recall previous phase I/II studies describing its role as an efficient imaging tool for HER2-expressing breast cancer [78]. Moreover, a second candidate, ABY-062, is about to be proposed for clinical studies as an inhalable agent that targets the cytokine thymic stromal lymphopoietin (TSLP) for treating asthma [79]. This scenario paved the way for a new concept of alternative therapeutics, which comes alongside those related to Abs. Indeed, though Abs have long been regarded as “magic bullets” in human therapies, they still suffer from undesirable properties, e.g., the large size that may hinder the penetration into tissues and the presence of the F_c_ regions that might give rise to antibody-dependent enhancement (ADE) upon viral infection. In addition, many Abs must be produced in mammalian cells, need posttranslational modifications, and often require difficult manipulation for drug conjugation via conventional chemical strategies [5]. However, it is worth recalling that new and efficient strategies based on so-called “antibody fragments” such as Fabs, scFvs, and nanobodies, are rising as valid alternatives for their rapid and better diffusion in tumor tissues because of their small size and binding properties, e.g., valency, thus reducing the barrier effect usually observed with whole Abs. Nevertheless, in many cases, these fragments are still accompanied by rapid clearance from the body, which may be beneficial for imaging purposes but detrimental in cases of therapies [80]. Therefore, it can be speculated that the paradigm of Abs in cancer therapies might become overtaken by affibodies and DARPins, due to their small size, very stable chemical behavior, and simpler architecture [81]. Nevertheless, the forefront of affibody- and DARPin-based NPs is still lacking approved trials for human uses though the number of published examples is increasing worldwide. In this scenario, two main properties must be recalled that highlight the putative advantages of these hybrids in clinical uses. First, from the formulation point of view, nanosized objects usually overcome the limitations related to poor solubility, low bioavailability, and many unfavorable existing chemotherapeutic agents [82]. Secondly, Abs-mimicking small molecules, especially affibodies, show high biodistribution and greater tissue penetration that are difficult to achieve. Many supporting studies are worth mentioning in this sense, for instance, those by Tolmachev et al. who showed rapid biodistribution and localized accumulation of affibodies in xenografted mice for imaging of EGFR-expressing tissues [83] or by Da Pieve et al. who produced a radio-labeled affibody able to specifically reach, bind, and image HER3, representing a challenging target marker modestly expressed by carcinoma cells [84]. More information about the excellent pharmacokinetic properties of affibodies can be found elsewhere [9]. In conclusion, one can fairly speculate that these new hybrid composites will quickly come under the limelight of cancer-related nanomedicine in the footsteps of nanomaterials functionalized with full Abs [85,86,87].

## Figures and Tables

**Figure 1 ijms-24-08680-f001:**
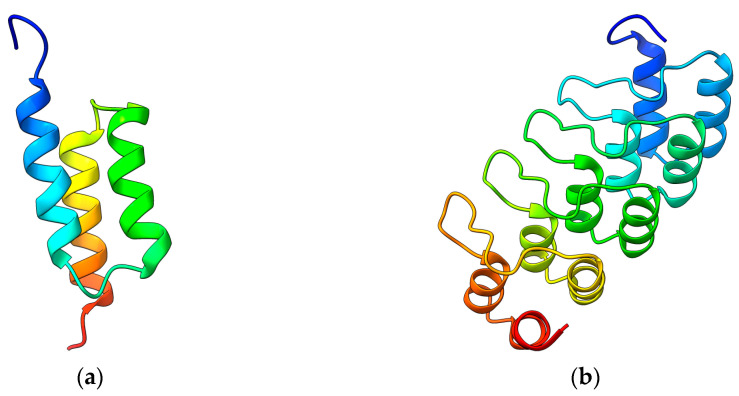
(**a**) Nuclear magnetic resonance structure of a Z affibody (PDB 2KZJ) [30]. (**b**) Crystal structure of a DARPin (2QYJ) [31]. Images obtained with ChimeraX v1.4.

**Figure 2 ijms-24-08680-f002:**
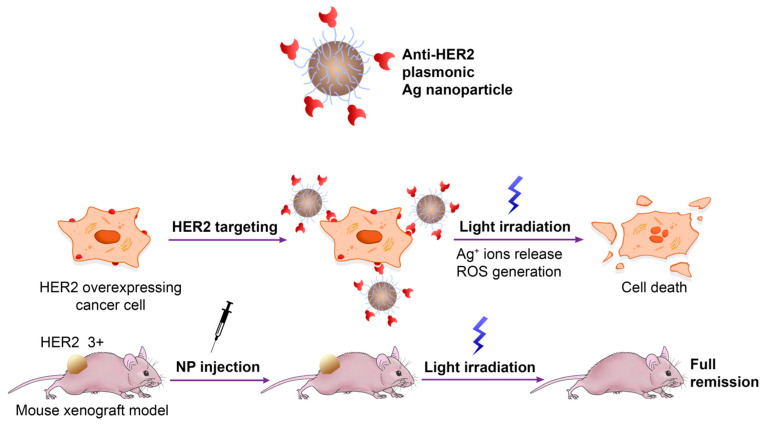
Plasmonic Z_HER2_-functionalized AgNPs for PTT cancer therapy. Adapted with permission from Ref. [41]. 2022, MDPI, Basel, Switzerland.

**Figure 3 ijms-24-08680-f003:**
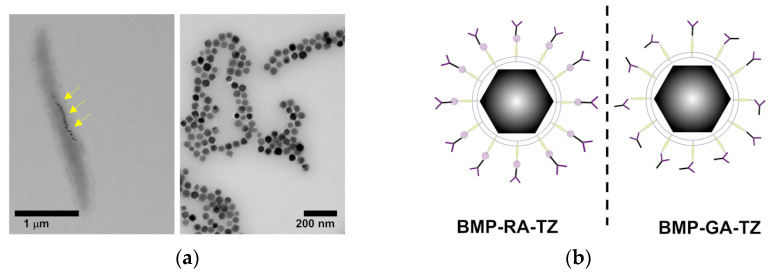
(**a**) Electron micrographs of *M. gryphiswaldense* containing BMPs before and after purification. Yellow arrows indicate the BMPs inside the organism. (**b**) Orientation of TZ antibodies on BMPs in the presence (**left**) or absence (**right**) of Z_HER2_ affibodies. Adapted with permission from Ref. [48]. 2022, Dove Medical Press Limited.

**Figure 4 ijms-24-08680-f004:**
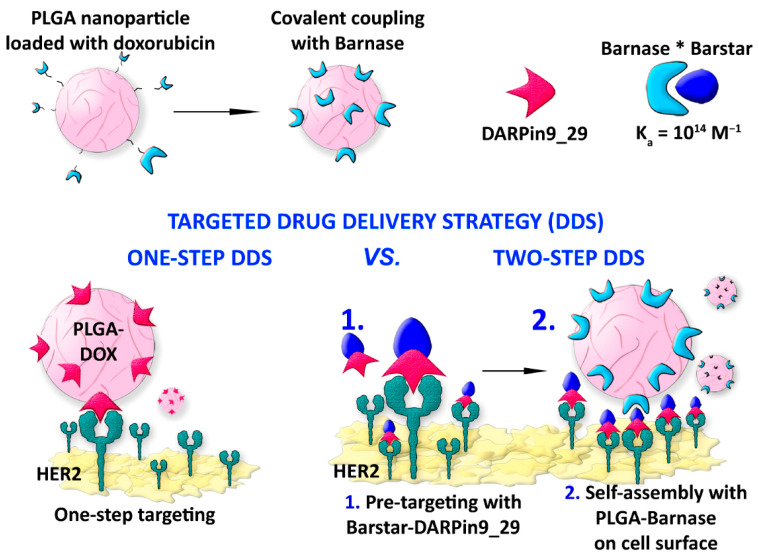
DARPin-assisted one-step and two-step drug delivery system using polymeric NPs. The strong barnase-barstar complex (barnase*barstar) is exploited to target cells. Adapted with permission from Ref. [50]. 2022, MDPI, Basel, Switzerland.

**Figure 5 ijms-24-08680-f005:**
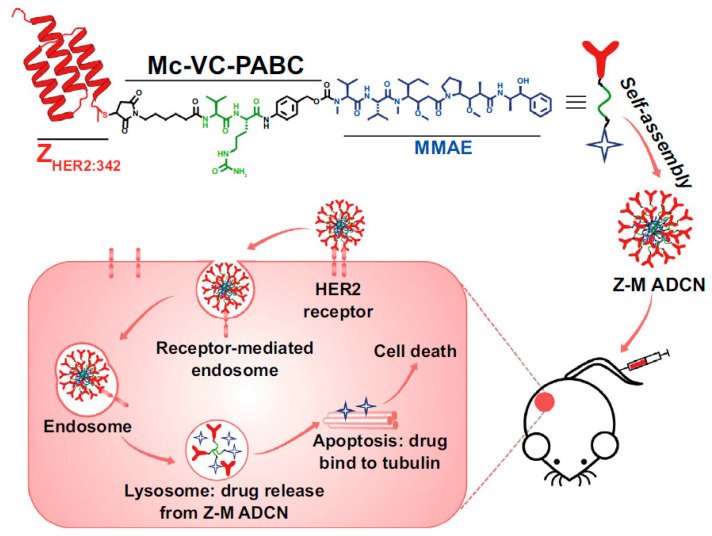
Self-assembled nanomicellar constructs targeting HER2 through affibodies. The construct enables long circulation time in blood and MMAE drug accumulation in tumors. Adapted with permission from Ref. [53]. 2023, Springer Nature Switzerland AG.

**Figure 6 ijms-24-08680-f006:**
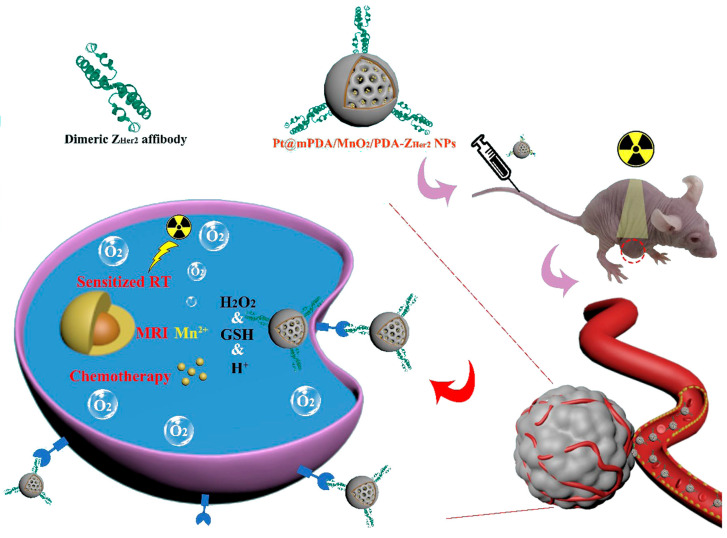
Cisplatin delivery on Z_HER2_–PDA–MnO_2_–PDANPs. The MnO_2_ nanomaterial acts as a layer to coat the constructs and increase biocompatibility in vivo. Adapted with permission from Ref. [59]. 2023, BioMed Central Ltd.

**Figure 7 ijms-24-08680-f007:**
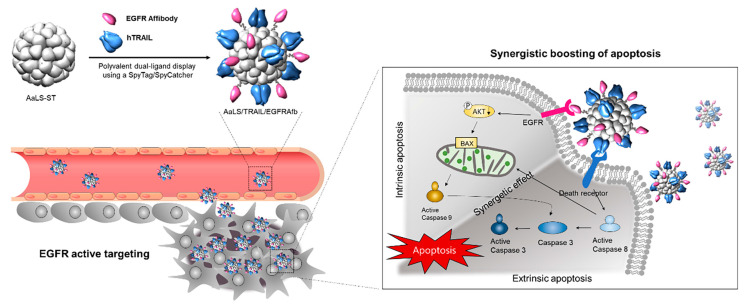
TRAIL- and Z_EGFR_-functionalized icosahedral Ls protein capsids for simultaneous, dual targeting of cancer cells and synergistic boosting of the apoptosis effects. Adapted with permission from Ref. [70]. 2022, Elsevier.

**Figure 8 ijms-24-08680-f008:**
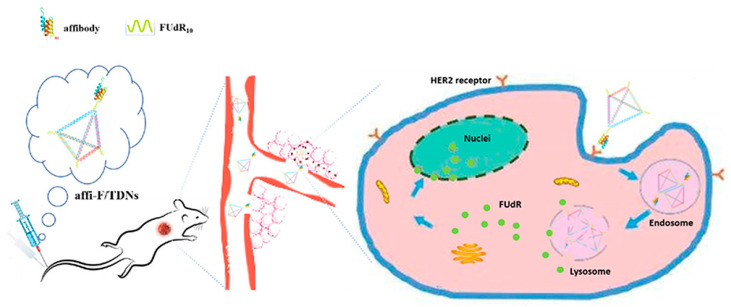
Z_HER2_-DNA nanotetrahedrons for drug delivery and loading of FUdR to enhance blood circulation and antitumor efficacy. Adapted with permission from Ref. [72]. 2020, Dove Medical Press Limited.

**Table 1 ijms-24-08680-t001:** Affibody- and DARPin-conjugated 0D nanostructures and their use in cancer therapy.

Inorganic Nanomaterials			
Construct	Therapeutic Effect	Target	Reference
Z_HER2_–AgNPs	Light-induced thermosensitizer	CHO and SKOV3-1ip cells; BALB/c Nu/Nu mice	[41]
DARPins–AuNRs	Light-induced thermosensitizer	BT-474 and MDA-MB-231 cells; BALB/c nude mice	[42]
Z_HER2_–AgNPs	Light-induced radiosensitizer	SK-BR-3, HN-5 and SKOV3 cells	[43]
Z_HER2_–DNA–AuNPs	FUdR and Dox-loaded drug carrier	BT474 and MCF-7 cells	[44]
Z_EGFR_–CD–UCNPs	5-FC prodrug converter	Caco-2 cells; BALB/c athymic nude mice	[45]
Z_HER2_–PbSQDs	ZnPP-loaded drug carrier	SKBR3 cells	[47]
Z_HER2_–BMPs	TZ-loaded drug carrier	SKBR3 cells	[48]
**Organic Nanomaterials**			
**Construct**	**Therapeutic Effect**	**Target**	**Reference**
Z_HER2_–PLGANPs	Rose Bengal-loaded drug carrier	CHO and SK-BR-3 cells	[51]
Z_HER2_–RALANPs	FUdR-loaded drug carrier	N87 cells	[52]
Z_HER2_–MMAE nanomicelles	MMAE-loaded drug-carrier	SKOV-3, BT474 and MDA-MB-231 cells; tumor-bearing mice	[53]
Z_PDGFRβ_–MMAE nanomicelles	MMAE-loaded drug carrier	COLO 205, HCT-116 and LS174T cells; BALB/c nude mice and SD rats	[54]
DARPin–barstar and PLGANPs–barnase	Dox-loaded drug carrier	SK-BR-3 and CHO cells	[50]
Z_HER2_–Dox–PLGANPs; DARPin–LoPE	Dox- and LoPE loaded drug carriers	SK-BR-3, BT-474, A549 and CHO cells; BALB/c nude mice	[55]
**Hybrid Nanomaterials**			
**Construct**	**Therapeutic Effect**	**Target**	**Reference**
Z_HER2_–cisplatin–PDA–MnO_2_–PDANPs	Cisplatin-loaded drug carrier and Light-induced radiosensitizer	SKOV-3 and MCF-7 cells; BALB/c nude mice	[59]
DARPins–Fe_3_O_4_NPs–polyarginine–dextranNPs	Ultrasound-induced Dox-loaded drug carrier	SKOV-3 cells;BALB/c nude mice	[60]
**Biological Nanomaterials**			
**Construct**	**Therapeutic Effect**	**Target**	**Reference**
Z_EGFR_–liposomes	Dox-loaded drug carrier	A431 and B16F10 cells; BALB/c (nu/nu) nude mice	[66]
Z_EGFR_–liposomes	Cisplatin-loaded drug carrier	A431 cells; BALB/c (nu/nu) nude mice	[67]
DARPins–liposomes	Ribonuclease barnase-loaded toxin carrier	SK-BR-3 cells	[69]
Z_EGFR_–*Aa*LSNPs	TRAIL-loaded drug carrier	A431 cells; BALB/c (nu/nu) nude mice	[70]
Z_EGFR_–DNA	FUdR-loaded drug carrier	BT474;NOD/SCID mice	[72]
Z_HER2_–DNA	FUdR- and Cur-loaded drug carrier	N87 cells	[73]

## Data Availability

Not applicable.

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
