# Peer review of "Recent Advances on Affibody- and DARPin-Conjugated Nanomaterials in Cancer Therapy"

_ijms, 2023, doi:10.3390/ijms24108680_

Round 1

Reviewer 1 Report

1-In the abstract, you mentioned to thermal stability of both affibody molecules and DARpins. Line 13: ‘’ strong binding affinity, solubility, chemical and thermal stability, small size, multiple’’

As far as I know and I have worked with both, we can claim that affibodies are thermally and chemically stable but some of DARpins cannot tolerate temperature over 60℃. I suggest to correct above sentence.

2-In line 285, you mentioned to PLGANPs for the first time in the paper. Please provide the full name for this abbreviation.

3- In the first paragraph of discussion part, I recommend to use of anti-HER2 ABY-025 affibody molecule for imaging of HER2-expressing breast cancer in patients as a good reference for clinical application of Affibody molecules. (J. Sörensen, et al., Theranostics 2016, Vol. 6, 262-271.)

1-Quality of English Language is Ok but many sentences must be corrected grammatically.

2- The use of shorter sentences is recommended.

Author Response

Dear Reviewer,

please find below our point-by-point reply.

1. In the abstract, you mentioned to thermal stability of both affibody molecules and DARpins. Line 13: ‘’ strong binding affinity, solubility, chemical and thermal stability, small size, multiple’’

As far as I know and I have worked with both, we can claim that affibodies are thermally and chemically stable but some of DARpins cannot tolerate temperature over 60℃. I suggest to correct above sentence.

We thank the Reviewer for this hint. The sentence has been modified acordingly.

2. In line 285, you mentioned to PLGANPs for the first time in the paper. Please provide the full name for this abbreviation.

The full name has been added and a list of abbreviations provided at page 21.

3. In the first paragraph of discussion part, I recommend to use of anti-HER2 ABY-025 affibody molecule for imaging of HER2-expressing breast cancer in patients as a good reference for clinical application of Affibody molecules. (J. Sörensen, et al., Theranostics 2016, Vol. 6, 262-271.)

We thank the Reviewer for this hint. A sentence reporting on the suggested study has been added and the proper credit provided at page 19, line 623.

4. Quality of English Language is Ok but many sentences must be corrected grammatically.

The text has been revised to correct and improve the readibility of many sentences when appropriate.

5. The use of shorter sentences is recommended.

The text has been revised to shorten many sentences when appropriate.

Reviewer 2 Report

Thank you for giving me the opportunity to read your work. Please find attached my comments.

No comments.

Author Response

Dear Reviewer,

please find below our point-by-point reply.

1. I found figures and tables very helpful to follow the text. However, I miss a section where conjugations strategies are commented more in detail. These strategies are summarized in table 2 and commented briefly on some parts of the text, but I believe a proper section would be of high interest.

We agree with the Reviewer. A section has been added at page 16 describing the main strategies used to realize these hybrid constructs and provided new references. Please, note that the mentioned strategies exclusively refer to the studies mentioned in the review.

2. Is there any information regarding the number of affibodies or DARP ligands per nanoparticle?

We found a few comparative studies reporting on the number of affibodies and DARPins linked to spherical NPs and the correlation with the targeting efficacy. Moreover, these studies also compare these hybrids with antibodies-conjugated NPs. These results would further highlight the usefulness of affibodies and DARPin and therefore we added these information in the section of the conjugation statetgies at page 16, line 590 alongside with new references.

3. Can the authors comment briefly the implications of targeting with these molecules in pharmacokinetics of nanoparticles? Specially compared with antibodies or antibody fragments. Also a comparative of avidity, molecular weight or other issues would be interesting.

A comment about the effects of affibodies in pharmacokinetics of NPs has been added at page 20, line 648 in the Conclusions section and the proper references provided. For that concerning the avidity, a comment has been added in the section of Conjugation stretegies at page 17, line 597 and the related references provided accordingly.

4. Some information regarding clinical status of these targeted nanoparticles would be interesting.

We agree with the Reviewer. However, to the best of our knowledge no information about clinical studies are reported at now for affibodies- orDARPins-conjugated NPs.

4. A list of abbreviations would be of help.

We thank the Reviewer for this suggestion. A list of abbreviations has been added at page 21.

Reviewer 3 Report

1.       What is the basis of limiting the studies in the Review to the last two years? There is a number of significant studies in the field that are omitted, for example, by Guryev et al. describing the use of upconversion nanoparticles (UCNP) coupled to yttrium-90 and exotoxin A fused with DARPin targeting HER2. Guryev EL, Volodina NO, Shilyagina NY, Gudkov SV, Balalaeva IV, Volovetskiy AB, Lyubeshkin AV, Sen' AV, Ermilov SA, Vodeneev VA, Petrov RV, Zvyagin AV, Alferov ZI, Deyev SM. Radioactive (90Y) upconversion nanoparticles conjugated with recombinant targeted toxin for synergistic nanotheranostics of cancer. Proc Natl Acad Sci U S A. 2018 Sep 25;115(39):9690-9695. doi: 10.1073/pnas.1809258115. Epub 2018 Sep 7. PMID: 30194234; PMCID: PMC6166851. It could be recommended to add a number of studies significant to the field where the in vivo therapeutic efficacy was demonstrated.

2.       I recommend to structure The Tables 1 and 2 to improve their readability by using a number of parameters, e.g. by sorting the studies based either on the type of targeting molecule (first affibody, then DARPin) or by the type of drug used (Dox, LoPE) or similar.

1.       The use of more scientific language is recommended, e.g. page 2 line 72 “also nanomaterials have been “contaminated” with both affibodies and DARPins” the word “contaminated” should be replaced by a more appropriate term.

2.       Sometimes the phrases are overcomplicated and difficult to understand, e.g. Page 2 line 113 “These properties contributed to focus on affibodies a high scientific interest in terms of  applied science making them more appealing than Abs.” the part “contributed to focus on affibodies a high scientific interest in terms of applied science” – could be written more clearly.

3.       The use of shorter sentences is recommended, e.g. pages 2-3 lines 92-105 has two very long sentences which are difficult to read.

Author Response

Dear Reviewer,

please find below our point-by-point reply.

1. What is the basis of limiting the studies in the Review to the last two years? There is a number of significant studies in the field that are omitted, for example, by Guryev et al. describing the use of upconversion nanoparticles (UCNP) coupled to yttrium-90 and exotoxin A fused with DARPin targeting HER2. Guryev EL, Volodina NO, Shilyagina NY, Gudkov SV, Balalaeva IV, Volovetskiy AB, Lyubeshkin AV, Sen' AV, Ermilov SA, Vodeneev VA, Petrov RV, Zvyagin AV, Alferov ZI, Deyev SM. Radioactive (90Y) upconversion nanoparticles conjugated with recombinant targeted toxin for synergistic nanotheranostics of cancer. Proc Natl Acad Sci U S A. 2018 Sep 25;115(39):9690-9695. doi: 10.1073/pnas.1809258115. Epub 2018 Sep 7. PMID: 30194234; PMCID: PMC6166851. It could be recommended to add a number of studies significant to the field where the in vivo therapeutic efficacy was demonstrated.

We thank the Reviewer for suggesting this hint. We are aware that many interesting examples have been omitted by limiting the review to recent years. The choice was an arbitrary one and done in the attempt to provide a comprehensive but still concise overview at the same time. However, we understand the importance of recalling important studies related to dual therapeutic-imaging nanotools as the one suggested; therefore, we mentioned it and added the reference as a supporting example at page 6, line 244.

2. I recommend to structure The Tables 1 and 2 to improve their readability by using a number of parameters, e.g. by sorting the studies based either on the type of targeting molecule (first affibody, then DARPin) or by the type of drug used (Dox, LoPE) or similar.

Both tables have been rearranged by sorting according to the nanomaterial type (biological, organic, etc) reflecting the structure and order of the paragraphs.

3. The use of more scientific language is recommended, e.g. page 2 line 72 “also nanomaterials have been “contaminated” with both affibodies and DARPins” the word “contaminated” should be replaced by a more appropriate term.

The term has been replaced with “functionalized”. In general, other substitutions have been provided throughout the text when appropriate.

4. Sometimes the phrases are overcomplicated and difficult to understand, e.g. Page 2 line 113 “These properties contributed to focus on affibodies a high scientific interest in terms of applied science making them more appealing than Abs.” the part “contributed to focus on affibodies a high scientific interest in terms of applied science” – could be written more clearly.

The phrase has been rearranged. In general, the text has been revised to improve the readibility of the sentences when appropriate.

5. The use of shorter sentences is recommended, e.g. pages 2-3 lines 92-105 has two very long sentences which are difficult to read.

The sentence has been rearranged. In general, the text has been revised to shorten the sentences when appropriate.